# The Microbiome and Its Implications in Cancer Immunotherapy

**DOI:** 10.3390/molecules26010206

**Published:** 2021-01-03

**Authors:** Hani Choudhry

**Affiliations:** Department of Biochemistry, Faculty of Sciences, Cancer and Mutagenesis Unit, King Fahd Medical Research Center, King Abdulaziz University, Jeddah 21589, Saudi Arabia; hchaudahry@kau.edu.sa

**Keywords:** microbiome, cancer progression and microbes, physiological responses, CTLA-4, immunotherapy, immuno-oncology

## Abstract

Cancer is responsible for ~18 million deaths globally each year, representing a major cause of death. Several types of therapy strategies such as radiotherapy, chemotherapy and more recently immunotherapy, have been implemented in treating various types of cancer. Microbes have recently been found to be both directly and indirectly involved in cancer progression and regulation, and studies have provided novel and clear insights into the microbiome-mediated emergence of cancers. Scientists around the globe are striving hard to identify and characterize these microbes and the underlying mechanisms by which they promote or suppress various kinds of cancer. Microbes may influence immunotherapy by blocking various cell cycle checkpoints and the production of certain metabolites. Hence, there is an urgent need to better understand the role of these microbes in the promotion and suppression of cancer. The identification of microbes may help in the development of future diagnostic tools to cure cancers possibly associated with the microbiome. This review mainly focuses on various microbes and their association with different types of cancer, responses to immunotherapeutic modulation, physiological responses, and prebiotic and postbiotic effects.

## 1. Introduction

The microbiome refers to the total collection of trillions of microbes, such as bacteria, fungi, and viruses that are living in the human body. The presence of microbes can be beneficial or harmful to health. Microbes that are not pathogenic assist in digestion and the production of vitamins such as K, A, and E in addition to releasing certain enzymes and producing other useful byproducts such as acids. The major microbes are found in the intestinal tract and colon followed by skin and vagina [1]. These microbes have been evolved along with human beings as part of an evolutionary phenomenon characterized by processes lasting many years [2]. The development of the intestinal microbiome starts at a very early stage during childhood, through intimate interactions of the microbiome associated with the maternal body [3]. This ecology of the microbiomeis basically affected by the diet consumed by the mother, gestational age, delivery type, and exposure to various antibiotics [4]. Both types of innate and adaptive immune systems have intimate associations with the microbiome that continuously resides in the human intestine [5]. Recent studies have demonstrated that the intestinal microbiome plays very critical role in the regulation of human homeostasis and the immune system, and can probably causeshuman metabolic disorders, including cancer [6]. Studies have shown that commensal microbes are useful in supporting the immune system through inhibiting inflammatory responses and eliciting Toll-like receptors (TLR) responses [7]. The intestine is the site for the absorption of nutrients and other important metabolites. The metabolites of microbiota are also absorbed in the intestine and ultimately assimilate into the bloodstream. Once assimilated into the bloodstream, these metabolites start to associate with G-protein coupled receptors [8] and ultimately contribute to the regulation of human physiology and pathogenesis. Therefore, the gut microbiome is also called the human’s second brain. The GBA (gut–brain axis) works as a two-way flow of important information and signaling between the systems of the brain and the microbiome [9]. The gut microbiome has a great impact on the heart and its related diseases, the regulation of glucose metabolism, insulin sensitivity; andhuman energy homeostasis [10]. The microbiome is also associated with inflammatory pathways, lipid metabolism and the initiating obesity through genetic and epigenetic crosstalk [11]. The gut microbiota also affects mood fluctuations, especially in cases of depression and anxiety [12]. 10 × 10^6^.

The role of microbes in cancer initiation and progression have not been clearly ascertained yet. Some microbes have been broadly studied in certain cancer types, e.g., the presence of *F. bravibacterium* has been reported in colon cancer. *Firmicutes* and *Enterobacteriaceae* have been widely characterized in the case of gut cancer [13,14]. Different types of cancer have been reported to have different kinds of microbes (bacteria and viruses). Recent advances in metagenomics and transcriptomics analysis have made it easy to recognize and characterize various microbiomes in related cancer and their role in increasing or inducing physiological changes. While an individual’s microbiome in various types of cancer is supposed to be the same, environmental variations may lead to new kinds of microbes [15,16,17,18,19]. There is an urgent requirement to characterize microbes in healthy and diseased conditions. Correlations between these two conditions can help to trace the availability of microbes and the development of diagnostic tools for the targeted treatment of cancers. In this review, we summarize the microbes that are generally found to be associated with various types of cancer and the effect of immunotherapy on their microbiomes. In addition, the effect of probiotics and prebiotics on the role of microbes that play a role in disease and the associated physiological changes in the host and microbes will also be discussed.

## 2. The Microbiome and Its Association with Various Kinds of Cancer

The human gut has a diverse microbial ecosystem that includes viruses, fungi, archaea, and bacteria. Bacteria play a major role as microbiota and mostly belong to *phyla Bacteroidetes*, *Proteobacteria*, *Firmicutes*, and *Actinobacteria*. Microbes have been found to be involved and associated with 10–20% types of human cancer (Figure 1). Efforts are being made to characterize microbes affecting cancer progression and inhibition, including in affecting responses to various treatments given to patients. In response to cancer progression, various microbes have been reported that causes disease, as shown in Table 1 [20,21,22,23,24,25]. Increment level of *Escherichia coli*, *Staphylococcus bovis*, *Fusobacterium nucleatum*, *Clostridium* spp., *Streptococcus* spp., and *Bacteroides* have been reported incolorectal cancer, while decrement in level of *Lactobacillus*, *Microbacterium*, *Anoxybacillus*, and *Akkermansia muciniphila*. A list of common cancers and their associated microbes is given in Table 1 [26,27,28,29,30,31,32,33,34,35]. The presence of microbes could be used as a bioindicator to ensure early detection and therefore treatment of disease at its early stages. A method to trace those microbes in response to disease should be devised [36,37,38,39,40]. Microbes need to be characterized in each kind of cancer, and proper control measures need to be devised for treatment [40,41,42,43]. Several microbes have been causally linked to various types of cancers, e.g., *Helicobacter pylori* are the major causative agent of gastric lymphoma, gastric adenocarcinoma, and esophageal adenocarcinoma, while the Epstein–Barr virus (EBV) causes lymphomas and nasopharyngeal carcinoma [44,45]. Other viruses have been reported to cause other types of liver and lymphatic cancers are shown in Table 2 [46]. Intestinal flora and its metabolites work as major causativeagents ofcolorectal cancer. Diverse flora present in the intestine are manyprobiotic microbes, namely *Bifidobacterium*, *Streptococcus thermophilus*, *Lactobacillus rhamnosus* and *Lactobacillus acidophilus*, in addition to pathogenic bacteria such as *Enterococcus faecalis*, *Bacteroides fragilis*, *Clostridia*, *Fusobacterium nucleatum*, *Streptococcus bovis*, *Salmonella* and *Enterotoxigenic* [47,48]. Therefore, proper screening and microbiome classification could further provide knowledge of disease occurrence and progression.

Tumor promoting or suppressing effects have also been reported in gastric, lung, liver, and colorectal cancer when microbes are present [49]. Germ-free models have shown less capabilities of tumors growth in comparison with microbe-associated models, as shown in Table 3 [50,51,52,53,54,55,56,57,58,59,60]. Each type of microbe needs to be characterized so that the early detection of cancer is possible for every type of disease. Table 3 [50,51,52,53,54,55,56,57,58,59,60] shows the murine models for various types of cancers and the treatments that were applied, and their results in terms of reduced tumor incidence in relation to the presence of germs.

Microbe availability can influence physiological changes in cells. For instance, the presence of particular microbes can lead to inflammation via IL-10 generation. Recent studies on other types of cancer and microbial metabolomics have led to the discovery o new biomarkers. In colitis cases, there is a 10-fold increase in the incidence ofcolorectal cancer and inflammation occurs due to the presence of members of the *Enterobacteriaceae* family, such as *Enterococcus faecalis* and *Escherichia coli*. These strains are upregulated >100-fold in colon cancer [61,62,63].Polyketide synthases (*pks*) found in *E.coli*, synthesizes the genotoxiccolibactin. However, this gene is not found in *E. faecalis*, which makes *E. faecalis* less virulent in comparison to *E. coli*. The metabolic product of *E. coli*, *pks* induces DNA damage [63]. The risk factor may even be increased due to the presence of food, which affects the microbiota. The heterocyclic amines present in red meatare fermented by gut microbiota to yield hydrogen sulfide and electrophilic free radicals, which cause DNA damage and subsequent mutation via faulty DNA repair [64]. In cases of obesity, microbiota is found to be enriched by diverse microbes. Therefore, the chances of cancer occurrence are higher in obese persons [65]. Individuals on a high fat diet are rich in group IX *Firmicutes* bacteria, such as *Clostridium*, which convert bile acids into secondary compounds, often deoxycholic acid (DCA). DCA generates free radicals, so it is a potent carcinogen and has been reported to cause liver and colorectal cancers [66]. More information related to microbes and mechanisms is given in Table 4 [67,68,69,70,71,72,73,74,75,76,77,78,79,80,81,82,83,84,85,86,87,88,89,90,91,92,93,94].

### 2.1. Colon Cancer

Colon cancer is the third deadliest cancer globally. The colon is the main area where microbesrecides, especially bacteria (phylum *Firmicutes*). These bacteria remain inside the colon and utilize the food materials found within [95]. In response to food materials and their post metabolism, they secrete several byproducts into the bloodstream. These byproducts can increase or decrease the levels and expression of certain cancer-causing molecules and genes. Microbes promote intestinal homeostasis and anti-oncogenic responses but may elicit oncogenic responses through chronic dysregulated inflammation and genotoxic effects. Certain microbes that can increase the risk of colorectal cancer have recently been identified [95,96]. The promotion of a healthy vs. cancerous colon depends on the basic composition of the gut microbiota and the dynamic equilibrium within the microbial community [97,98]. Recently, disturbances in the normal population of microbiota have been reported to be the main cause of colorectal cancer (CRC), as shown by dysbiosis [99,100]. *F. nucleatum* and *E. coli* have been reported in most CRC patients [101]. Recent studies on mice have concluded that *F. nucleatum* directly participates in enhancement of tumor growth. Hence, *F. nucleatum* can work as a major prognostic biomarker in colon cancer studies [102] (Figure 2). CD3^+^ T cell count decreases as *F. nucleatum* increases. Proteins from *F. nucleatum* bind with TIGIT, E-cadherin, Fap2, and FadA and providing very important information for the design of drugs against these targets. *E. coli* is more tumorigenic with polyketide synthase than without, indicating that intestinal inflammation helps in targeting the cancer-inducing activity of the microbiota [103] (Figure 2). Butyrate, an important byproduct metabolite of SCFA (short chain fatty acid), is derived from *E. coli* and is a potential major genotoxic and epigenetic modulating molecule. It has deleterious effects on local gut flora. It is clear that balance of microbes present in the gut determines initiation and suppression of cancer. It has been shown that, compared with control mice, germ-free mice that lack microbes are more prone to tumor development after treatment with the carcinogens dextran sulfate sodium (DSS) and azoxymethane (AOM) [104].The mechanisms behind this support the repair of epithelial and barrier functions that allow for the resolution of inflammation at the time of epithelium damage.The down regulation of inflammatory pathways likely induces the prevention of dysbiosis, tumorigenesis and the increased apoptosis of tumor cells, while the up regulation of cytokines such as IL-18 likely induces antitumor responses and tissue repair.

### 2.2. Breast Cancer

Breast cancer is reported in one of eight women in the United States. It is the most prevalent cancer in women after lung and colon cancer. Breast cancer is regarded as a familial (genetically) HR^+^-related mutation and is linked with changes related to estrogen. However, in recent studies, it has been reported that microbes (especially in the gut) also play a role in progression in breast cancer. Bile acid production is controlled by microbes found in the gut. Dihydroxy acid (DHA), secreted by the action of microbes, especially *E. coli*, has a direct influence on the estrogen metabolome [105]. Excess estrogen can deregulate various regulated pathways and cause overexpression of factors that cause breast cancer. It is now known that microbes present in the gut, mouth, breast, and breast milk influence estrogen metabolism, inflammation, and epigenetic alterations [106]. Some bacteria and their secreted metabolites can affect different types of signaling pathways, such as E-cadherin/β-catenin [107], which functions as a major promoter of apoptosis and double-strand breaks in DNA [59], alters cell differentiation [108], and is linked with innate immunity and some toll-like receptors (TLRs), which activates inflammatory signaling pathways that ultimately help in maintaining body homeostasis [109,110,111]. In the case of Breast Cancer (BC), *Proteobacteria* and *Firmicutes* species have been found in greater abundance. In separate samples, these two groups of bacteria serve as the key causative agent of breast cancer. In patients with advance breast cancer, 16S rRNA analysis demonstrated a very high relative abundance of three groups of bacteria, namely *Staphylococcus*, *Bacillus*, and *Enterobacteriaceae* spp. Moreover, *Escherichia coli* (*Enterobacteriaceae*) and *Staphylococcus epidermidis* are known to be responsible for double-stranded DNA breaks [112]. A study on Irish and Canadian women with breast cancer found they were more likely to have elevated levels of bacteria belong to the *Enterobacteriaceae*, *Staphylococcus*, and *Bacillus* compared with women who do not have breast cancer [113]. *Enterobacteriaceae* (8.3%), *Prevotella* (5.0%), *Bacillus* (11.4%), *Acinetobacter* (10.0%), *Pseudomonas* (6.5%), *Propionibacterium* (5.8%), Comamonadaceae (5.7%), *Gammaproteo bacteria* (5.0%) and *Staphylococcus* (6.5%) were all represented as indicated in Canadian women with breast cancer. The most abundant represented taxon was *Pseudomonas* (5.3%), *Enterobacteriaceae* (30.8%), *Propionibacterium* (10.1%), *Listeriawelshimeri* (12.1%), and *Staphylococcus* (12.7%) were present in samples taken from Irish women with BC. The composition of bacterial and viral populations associated with breast cancer differs according to the examined breast position, as revealed by 16S RNA sample analysis, as explained by Zahra Eslami-S et al., 2020 (Figure 3). More abundant *Methylobacterium*, *Staphylococcus*and *Actinomycetes*were found in urine samples of breast cancer patients. *Lactobacillus acidophilus*, a well-known probiotic present in kimchi and yogurt, can easily enter the mammary gland of the breast and displays a variety of anticancer properties [114]. Women can face a high incidence of protective antioxidant effects if they consume fermented milk products. It is very widely stated that the abundance of *Lactococcus* and *Lactobacillus* spp. in healthy breast tissues, relative to cancerous breast cancer tissues, possibly plays a role in the cure and prevention of breast cancer. Some studies indicate that *Lactobacillus* is often involved in controlling the immune system and reducing the amount of C-reactive protein and IL-6 that function as pro-inflammatory factors during inflammatory reactions [115]. In benign and malignant breast tissue samples, there is an enormous difference in the persisting microbiome composition (Figure 3). A review of taxonomic studies reveals that in the case of benign and invasive breast cancer, the overall microbiota of breast tissue, were very similar and dominated by the bacterial phyla *Firmicutes* and *Bacteroidetes*, as expected. The malignancy associated with enrichment could be determined by the assessment of differential taxa of bacterial species between these two classes. The lower abundance taxa are likely to include *Hydrogenophaga*, *Lactobacillus*, *Atopobium*, *Gluconacetobacter* and *Fusobacterium* genera [116]. In the case of breast cancer, the carcinogenic effect of 10 well-known infectious pathogens has been identified. A greater understanding of the effects and function of these microbial agents in breast cancer would broaden our ability to prevent development of tumor and potentially lead to future diagnosis tools and treatments. In the late nineties, gastric cancer and breast cancer were linked to infections with *Helicobacter pylori* and *Salmonella typhi* these two well-known bacterium also participates in gallbladder cancer. More surprisingly, in the case of bladder cancer, breast cancer, colon cancer, and melanoma, *S. typhus* also serves as a promising carrier of therapeutic agents [116]. It is important to study the interactions between microenvironments and microbiomes linked to different organs within the human body and understanding their breast cancer development [117].

### 2.3. Oral Cancer

High throughput16S rRNA sequencing of mouth cavity of healthy individuals reveals the presence of five major phyla namely *Fusobacteria*, *Bacteroidetes*, *Actinobacteria*, *Proteobacteria*, and *Firmicutes.* Major genera *Veillonella*, *Streptococcus*, *Prevotella*, *Haemophilus*, *Neisseria* and *Leptotrichia* as found when assaying various parts of the mouth [118]. According to recent reports, the principal pathogen present in periodontal *Porphyromonas gingivalis* has been identified as a major biomarker for oral digestive cancer and related death and it also expand to colorectal and pancreatic cancer [119]. Recent studies suggest correlations between both oral fungal and viral microbes in the development of oral cancer. A prime example of this is *human papillomavirus 16* (HPV-16), a major causative agent of several carcinomas related to oropharyngeal squamous cells [120,121]. Early studies through culture-dependent assays show that squamous cells of oral carcinomas have enormously increased abundance of both aerobic and anaerobic types of bacteria. The anaerobes belong to *Clostridium*, *Veillonella*, *Fusobacterium*, *Porphyromonas*, *Prevotella* and *Actinomyces*, while aerobes belong to *Streptococcus*, *Haemophilus* and *Enterobacteriaceae*. In addition, about 30% of oral cancers have been shown to be due to *Candida albicans* [122].

### 2.4. Liver Cancer

The third leading cause of cancer mortality belongs to hepatocellular carcinoma (HCC). Bacterial dysbiosis, leaky gut, microbe-associated bacteria and metabolite molecular patterns work as key pathways that impel cancer-induction, genotoxicity, liver inflammation, and fibrosis [123,124]. Therefore, it is difficult to both manage and diagnose liver cancer. The microbiome present inside the human gut may answersome present unexpected questions. Researchers have developed a new method for microbiome-based identification of liver fibrosis and cirrhosis easily, economically with more than 90% accuracy [125,126]. Non-alcoholic fatty liver disease (NAFLD) is the world’s leading cause of chronic liver disease and can lead to cirrhosis and liver fibrosis and, eventually the development of cancer [127,128]. However, advanced diagnostic methods for liver cirrhosis and fibrosis diagnosis are still lacking. In certain regions of the liver, biopsies function as an invasive tool and can cause damage. Magnetic Resonance Imaging (MRIs) are costly and are less commonly available in rural areas [129]. One study group studied the microbiome as a method to administer new tests to classify patients as vulnerable to liver cancer in order to bypass potential development. This procedure is based on 19 species of bacteria that have been described and are often present in a patient’s stool samples. Microbiome signature associated with liver cirrhosis was identified using microbiome genetic profiling and metabolites analysis fromstool samples with 94% accuracy [130]. In future, this can potentially assist clinicians in assessing the stage and level of the disease and eventually promote the development care strategies. The basic patterns we observed reflect the complexity of the microbiome and the way in which liver cancer disease affects gut health [131]. The study suggested investigating the causal relation between the liver and microbiome disease. They are also optimistic that this method may be used to characterize other diseases, such as inflammatory bowel disease and Alzheimer’s disease, in which a dysregulated microbiome has been associated diseases [132,133]. Compared with minimal fibrosis and stable controls, the microbiome composition is very distinct in liver cancer patients with advanced fibrosis. The effect of the intestinal microbiome on the characteristics of chronic liver disease, its association with non-alcoholic liver fatty liver disease (NAFLD), and primary sclerosing cholangitis has recently been demonstrated [134].

Since advanced fibrosis is a significant predictor of liver disease-related mortality and morbidity, researchers are simply trying to find and classify biomarkers of gut microbiome fibrosis. Scientists have identified that NAFLD, hepatitis B, hepatitis C, and alcoholic liver disease are primarily caused by unique patterns in the gut microbiome of patients with cirrhosis [135]. Lower concentrations of *Bacteroides* and higher levels of *Prevotella* bacteria have been characterized by those with advanced fibrosis those of the species *Prevotellacopri*, which acts as a good indicator of advanced liver fibrosis and chronic liver disease. The components and relative abundance of the gut microbiome differ significantly between individuals. The number of distinct bacterial phyla, *Fusobacteria*, *Firmicutes*, *Proteobacteria*, *Actinobacteria*, and *Bacteroidetes* typically prevail in liver cancer [136]. For example, human *papillomavirus* infection, *H. pylori* and hepatitis C viruses are major risk factors for their respective development of cervical, stomach, and liver cancer [137]. *Salmonella enteric*, which causes damage to DNA, produces secondary bile acids that support inflammation and contribute to tumor development [138]. *Escherichia coli* cause DNA damage to hydrogen sulfide production, which can decrease mucous production and contribute to intestinal barrier breakdown [139]. The development of reactive oxygen species, lipopolysaccharides, and reactive nitrogen species is indicated by *Fusobacterium nucleatum*, which shows leaky junctions and inflammation [140]. This is likely to be due to the complex interplay of a variety of other variables, such as human biology, exposure to contaminants or other pollutants in the atmosphere, and lifestyle (diet, drinking and smoking) [141]. Among these variables, the impact of dieting on the gut microbiome is best understood and well founded. The key cause of the development of potentially carcinogenic compounds has been found to be obesity and calorie-rich diets high in protein and fat. N-nitroso derivatives, secondary bile acids, and branched-chain fatty acids are primarily found in these derivatives. Decreased cancer risk has been associated with diets that are high in fiber and plant compounds such as glucosinolates (sulfur-containing compounds), polyphenols, and flavonoids [142]. *Bifidobacterium longum* is a gut commensal short-chain fatty acid (SCFA)-producing bacterium that helps in preventing cancer. SCFAs specifically involve butyrate, propionate, acetate [143], and complex carbohydrate and dietary fiber bacterial fermentation throughout the gut. These compounds can mainly assist in maintaining close bowel junctions and nourishing colon cells. Other beneficial bacteria include *Lactobacillus acidophilus* which helps preserve DNA [144], and *Saccharomyces boulardii,* which can minimize inflammation by preventing DNA damage and tumor growth [145,146,147].

## 3. Microbiome Impact on Immunotherapy

Under normal circumstance, the immune system work to defend the host against infectious diseases, autoimmunity, and allergy through the action of a series of co-stimulatory and co-inhibitory receptors and their ligands, commonly called immune checkpoints [148]. According to recent evidence, tumors use many of these pathways to evade antitumor immune responses and eventually progress, disseminate, and metastasize. Immune checkpoints are molecules that need to be stimulated or inactivated by specific immune cells to initiate a specific form of immune response to fight disease or inflammation [149]. Many cancer cells often avoid being targeted and destroyed by the immune system via these checkpoints. As shown by monoclonal antibodies (mAbs) inhibiting cytotoxic T-lymphocyte antigen-4 (CTLA-4), and programmed cell death protein1/programmed cell death ligand 1 (PD-1/PD-L1) [150,151], immune checkpoint inhibitors (ICIs), referred to as novel immunotherapeutic agents, have shown very promising clinical implications for advanced hematologic malignancies. It is well-established that microbiome changes to the tumor microenvironment (TME) result in the advancement of immunomodulatory effects [152]. However, these implications pose new and wide-range concerns, such as whether there is a correlation between cancer immunotherapy and the gut microbiome. Here, we concentrate on recent data on developments in cancer immunotherapy and intestinal microbiome to address these issues. Drugs that are widely used to target these targets are providing a positive road to cancer treatment. These medications are well-known as checkpoint inhibitors [150]. Cancer immunotherapy has become a very exciting and evolving field of modern oncology for treating cancer patients. To generate an antitumor effect, the immune system uses these control points and their inhibitors. The role of microbiota in managing the immune system and inflammatory reactions has been demonstrated using germ-free mice. Inappropriate inflammatory reactions against commensal microbiota such as *E. coli* and *E.hirae* are prevented by interleukin-10 (IL-10) [153]. *E. faecalis* IL-10 knockout mice have shown a form of colitis phenotype. However, if grown in a germ-free environment, the colitis activity of knockout mice can be suppressed [154]. Growth factor-β1 knockout mice also carry these forms of cancer in a germ-free background [155]. To establish the role of microbes in certain types of humandiseases through the provision of specific microbes, the use of germ-free mice is important. Based on previous research, multiple findings indicate that intestinal microbes influence antitumor activity in a variety of ways. Microbes or their metabolic products, through their association with antigen-presenting cells (APCs) or toll-like receptors (TLRS) help to alter the immune response [156]. In reaction to chemotherapy, preclinical models have been used to demonstrate the impact of microbiota on the gut. Cyclophosphamide and oxaliplatin have been shown to modulate antitumor activity by involving local microbes and their metabolites (maturation of T helper 17-TH 17 is achieved) for chemotherapy purposes in cancer treatment [157].

Specific gut microbiota affects the immune response to different treatments during the immunotherapy, surgery, and radiation [158]. Further research is needed to evaluate the increase or decrease in the number of microbes during different treatments strategies provided for cancer treatment. Conclusions on the impact of microbes in relation to cancer therapy including the studied models, are detailed in Table 5 [159]. Fecal microbiota transplantation (FMT) is now being investigated with different cancer types and cell models and the findings with reference to immunotherapy are also shown in the referred Table 6 [160]. The relation between gut microbiome composition and the efficacy of carcinoma therapy is shown in Table 7. This table presents the key findings of preclinical and clinical studies showing the link between gut bacteria and the results of various treatments for different types of cancer and treatment regimes.

### 3.1. Microbiome Implications in CTLA-4 Based Immunotherapy

CTLA-4 is a type of protein found in specific T cells that function as a type of “off and on switch” that introduces vigilance into the immune system [175]. Ipilimumab (Yervoy) is a CTLA-4-associated monoclonal antibody (mAbs) that improves the body’s immune response to cancer cells. It is widely used for the treatment of skin melanoma and cureently under investigation for other forms of cancers [176]. Some microbes such as *B*. *fragilis*, *B*. *thetaiotaomicron*, *B.cepacia*, *G*. *formicilis*, and *F*. *prausnitzii* alter in response to the immune checkpoint inhibitor. *Mycobacterium bovis* has been used to treat bladder cancer for over a century, and other microbial products have been used for immune activation, apoptosis induction, and vasculogenesis inhibition [177,178]. Recent reports suggestedthat the microbiome plays a major role in improving immunotherapy, focusing mainly on PD-1 pathways and checkpoint inhibitor therapy targeting CTLA-4 [179]. Recent observations indicate that the response to checkpoint inhibitors differs greatly between patients. Responses, such as gastrointestinal and liver toxicity associated with checkpoint inhibitor therapy and differences in treatment-related toxicity may also be beneficial [180]. Some studies, for example, understanding the role of *Akkermansia muciniphila*, a gut microbiome enriched with *Bacteroidetes*, helps to answer these questions [181]. *Bifidobacteria* spp. Are known to protect against anti-CTLA-4-associated immune-mediated colitis [182]. Other studies have shown that it is safer to react to PD-1 blockades for healthy bacteria in the intestinal microbiome [183]. In patients with *Faecalibacterium prausnitzii*, higher loads and lower abundance of *Bacteroides* after anti-CTLA-4 therapy led higher risk factors for colitis [184]. *Bifidobacteria*, *Akkermansia muciniphila*, and *Ruminococcaceae* bacteria are usually associated with health factors. Those related to immunogenicity are *Alistipes*, *Collinsella* and *Enterococci*etc alters anti-CTLA-4 therapy [185]. The risk of infection and graft-versus-host disease (GVHD) after allogeneic hematopoietic stem cell transplantation (ASCT) in hematological malignancies is controlled by gut bacteria. Early applications of systemic broad-spectrum antibiotics are correlated with increased GVHD and mortality associated with transplantation, probably due to the depletion of the gut microbiota of defensive *Clostridiales* and *Blautia* species [186]. CTLA-4 Ab injections are sufficiently important to influence the content of microbiomes at the level of the genus. Studies on *Burkholderiales* and *Bacteroidales* found that the CTLA-4 blockade rapidly causes a comparative increase in *Clostridiales* quantity in feces [187]. Quantitative polymerase chain reaction (qPCR) studies have shown that targeting the *Bacteroides* genus and bacterial species in small intestine mucosa and feces are novel trend to study [188]. Promenent increment of content observed in organisms such as *B*. *uniformis* and *B*. *thetaiotaomicron* (Bt) from 24 to 48 h after CTLA-4 Ab injection in the mucosa of the small intestine [189]. Isolated and distinguished by one of the main regulatory *Bacteroides*, *B*. *fragilis* (Bf) was quickly found to be observable by regular colon mucosal PCR, while no increment of CTLA-4 Ab observed [190]. The therapeutic effects of cyclophosphamide administration [191] are decreased by some particular bacterial organisms, such as *Parabacteroides distasonis*, which drives Treg effects and SFBs that, in turn, drive Th17 responses. In response to cyclophosphamide, certain types of Gram-positive bacteria, such as *Enterococcus hirae* and *Lactobacillus johnsonii*, have been reported to increase [192] and carry Th1 memory cells into the lumen. Some bacterial organisms, such as *Clostridiales*, suppress immune cell responsiveness by activating the development of IL-10 in the intestine and extra-intestine and the differentiation of Tregs [193]. Commensal bacteria also aid in the regulation of systemic immunity in addition to influencing local immunity. B-created polysaccharide (PSA) *Bacteroides fragilis*is capable of reliably detecting immune defects in germ-free mice linked to Th1/Th2 imbalance and CD4^+^ T cell deficiency [194]. *E*. *hirae* activates the response of pathogenic Th17 (pTh17) cells and increases the extra-intestinal tissue ratio of cytotoxic T cells/Tregs, while *B*. *intestinihominis* bacteria helps to strengthen the response to systemic Tc1 and Th1 [195]. The gut microbiome is simultaneously formed and enriched by host immunity. In the case of a mouse model, adaptive immune and innate responses are downregulated with manybacterial enrichment [196]. Recent reports indicate that host immunity can also influence the morphology of certain species of bacteria which, in turn, often hampers the relationship between bacteria and epithelial cells (Figure 4) [151]. Cancer immunotherapy with anti-CTLA-4 antibodies modulates the balance of the microbiota–intestinal barrier by inducing IEC-mediated intestinal epithelial cell (IEL) apoptosis, resulting in disruption of the barrier. In experimental settings (possibly due to pathogenic Th17 cells (pTh17)), barrier perturbation is further increased during the co-blockade of IL-10 signaling or ICOS (Inducible T-cell COStimulator) signaling, resulting in higher intestinal toxicity resembling early signs of colitis [151]. (Re)colonization of mice treated with antibiotics by *B*. *fragilis* (Bf) and *Burkholderiacepacia* minimizes the toxicity caused by anti-CTLA-4 mAb (possibly through the plasmacytoid DC (pDC) mobilization ability of *B*. *fragilis*), which facilitates the proliferation of ICOS^+^ Treg in the lamina propria while retaining good antitumor efficacy (right). Increased bacterial species uptake, for example of *B*. *fragilis*, due to lamina propria DCs or to the possible DC absorption of soluble bacterial products results in the maturation of DCs and the development of IL-12, enabling T cells such as Th1 cells to be primed/activated (facilitated by the ongoing immune checkpoint blockade). These T cells, possibly cognizant of tumor antigens or cross-reactive bacterial antigens, are involved in antitumor immune responses (Figure 4) [197,198].

### 3.2. Microbiome Implications in PD-1/PD-L1 Inhibitor-Based Immunotherapy

PD-1 is a checkpoint protein found in immune T cells. It assists and preserves the protection of T cells from attacking and destroying other cells in the body [198]. There is a significant expression of PD-L1 in several cancer cells, which presumably allows them to evade an immune attack. Recent reports show that monoclonal antibodies targeting either PD-1 or PD-L1 can block this interaction and eventually increase the immune response to cancer cells [199]. In the treatment of variety of cancers, these drugs are a very promising approach. Generally, these medications are administered intravenously by physicians. Cemiplimab (Libtayo), pembrolizumab (Keytruda), and nivoluma are examples of drugs that aid in targeting PD-1 [200]. In the treatment of many forms of cancer, these medications were shown to be effective. New forms of cancer are being examined against these drugs give a ray of hope. Drugs that target PD-L1, namely, durvalumab (Imfinzi), (atezolizumab (Tecentriq), and avelumab (Bavencio) [201] are PD-L1 inhibitors. These medicines have already been shown to be effective in the treatment of different forms of cancer and are waiting to be tested for use against new cancers. Some inosine-related microbiomes have been reported to modulate the effectiveness of these inhibitors [202]. Three bacterial species have been shown to increase the effectiveness of immune checkpoint inhibitors, including *Bifidobacterium pseudolongum*, *Lactobacillusjohnsonii* and *Olsenella* [203]. Microbes may be used for the immunotherapy of cancer patients as a successful efficacy booster for PD-1 and PD-L1 inhibitorsCombinations of microbiomes that might boost the efficacy of immune-check inhibitors should be tested. Microbial adjuvants could be developed to increase the efficacy of inhibitors in the immunotherapy of cancers [204]. Advance research on the microbiome and its efficacy could promote the treatment of cancer on a personalized basis. Microbial distribution in healthy and disease subjects needs to be explored. Food items with enriched microbes responsible for increasing the efficacy of treatment can be provided. More human correlation studies are required to establish the identity of important microbes and their role in treatment (Figure 5) [205,206].

### 3.3. Microbiome Implications in Allo-HSCT (AHSCT)

Due to intense treatment with antibiotics, irradiation, and chemotherapy, some form of disruption in the intestinal microbiota contributes to the creation of rigorous gut graft-versus-host disease, often leading to serious infection. This often leads to worse results in patients undergoing allogeneic hematopoietic stem cell transplantation (allo-HSCT). The two main causative agents, *Ralstonia pickettii* and *Staphylococcus haemolyticus*, were found to be associated with a higher risk and mortality [207]. Allo-HSCT is a very effective approach for the treatment of different forms of hereditary hematopoietic disorders and associated hematological malignancies [208]. Allo-HSCT recipients must undergo intense complete body irradiation and chemotherapy [209] to exterminate latent malignant cells and immunocompetent cells.

In allo-HSCT, three major microbes, namely *Neisseria*, *Prevotella*, and *Streptococcus*, were found to be very high in quantity. The existence of these microbes can exacerbate the stem cells drawn from them. These microbes secrete certain metabolites such as short-chain fatty acids, which affect the incidence of certain diseases and graft rejection. A strong predictor of high risk of development and mortality associated with the introduction of allo-HSCT [210] has now been found based on the microbial diversity present in intestinal microbiota. Therefore, the graft transplant and its success rate are influenced by microbes. However, the precise mechanism of graft rejection through the intervention of microbes has yet to be identified.

### 3.4. Microbiome Implications in Probiotic Immunotherapy

Currently, there are very fewer therapeutic methods available to control and prevent treatment of underlying diseases such as liver, colorectal, oral, and breast cancer. Taking the importance of disease progression into account, the gut–microbiota disease axis has been shown to be a promising target to studythe development of cancertreatment.

#### 3.4.1. Liver Cancer

Recently, probiotics have been implemented for re-equilibrating the gut microbiome in the case of chronic liver disease (CLD) by selectively recruiting beneficial bacteria. However, recentreports haveshown the efficacy of probiotics in curing liver diseases both inanimal models and in patients [211]. In the case of rat models of DEN-mediated hepatocarcinogenesis, the administration of VSL#3 (specifically affecting *Lactobacillus paracasei*, *Lactobacillus acidophilus*, *Lactobacillus plantarum*, *Lactobacillus delbrueckii sub* sp. *Bulgaricus*, *Bifidobacterium infantis*, *Bifidobacterium breve*, *Bifidobacterium longum*, and *Streptococcus thermophiles*) was effective in enteric dysbiosis, decreasing inflammation of the intestine and leading to a decrease in liver tumor growth and its multiplicity [212].

#### 3.4.2. Breast Cancer (BC)

*Staphylococcus hominis* and *Enterococcus faecalis* have been shown to be inhibited in recent in vitro and in vivo studies, showing the impact of probiotics on breast cancer. The probiotic *Lactobacillus reuteri* suppresses early-stage cancer and has contributed to an increase in susceptibility to apoptosis in breast cells [213]. The anticancer effects of advanced probiotics on cancer cell lines have been studied by Mendoza et al. They identified the effects of cytotoxicity, the anti-proliferative action of cell cycle arrest, apoptosis, and probiotic effects. Another group (NCT03760653) described the impact of probiotics (*Bifidobacterium bifidum*, *Lactobacillus rhamnosus*, *Lactobacillus acidophilus* and *Lactobacillus paracasei*) and physical activity on bacterial equilibrium onthe immune system of BC survivors [214].

#### 3.4.3. Colorectal Cancer (CRC)

CRC can be inhibited by different kinds of probiotics througha variety of mechanisms. Recent studies have shown that probiotics lead to the biotransformation and detoxification in conjunction with carcinogenic mutagens, depending on the secreted glycoproteins, peptidoglycan, and polysaccharides on the surface of probiotics. In order to prevent CRC, probiotics can assist in the down regulation of inflammation and eventually reduce the levels of carcinogenic compounds and metabolites [215]. Recent studies have shown that, in mice treated with *Clostridium butyricum* and 1,2-two hydrazine hydrochloride, tumor size decreases due to decreased amounts of Th2 and Th17 cells, leading to the inhibition of CD4^+^ and CD8^+^ T lymphocytes. This led to decrease secretion of inflammatory factors such as nuclear factor B and IL-22, which jams cell cycles and induces apoptosis of tumor cells [216].

#### 3.4.4. Oral Cancer

Certain commercially available live probiotic strains of bacterial genera, such as *Streptococcus*, *Bifidobacterium* and *Lactobacillus* can increase the alpha diversity of the oral microbiome without altering its composition [217]. As described earlier, in terms of the gut microbiome, *Lactobacillus* is a widespread and effective probiotic [218], and has been found to be directly and indirectly associated with dental caries and CRC. *Streptococcus oralis* is implicated in the infection of cystic fibrosis, and *Leptotrichia* is implicated in pancreatic cancer. *Lactobacillus* is favorable to dental caries development without a microenvironment and helps promote better alpha diversity, leading to better oral health [219].

### 3.5. Microbiome Implications in CpG-Oligonucleotide (CpG ODN) Immunotherapy

Early pre-clinical and clinical studies indicate that unmethylated CG dinucleotide (CpG ODN) synthetic oligodeoxynucleotides have a powerful immunostimulatory property and can be used for a variety of treatments for cancer due to potent anticancer activity [220]. In a range of preclinical models, CpG ODNs and microbiome synergy have recently been implicated. Early clinical studies have shown that monoclonal antibodies and CpG ODNs can administered together safely [157]. Studies on preclinical models have shown that when combined with radiation therapy and chemotherapy, CpG ODNs can also improve antitumor activity. A type of pleiotropic cytokine formed by several cells is interleukin-10 (IL-10). IL-10 is well known for controlling the activity of APCs and helping to suppress the proinflammatory cytokine response by delaying T-cell activation [221].

In comparison, in the case of CD8 + TILs cells, tumor progression through mediationof IL-10 signaling can also be suppressed. Therefore, IL-10 also plays an important and paradoxical function in cancer immunotherapy via its potential to inhibit or activate IL-10 signaling [222]. Therefore, the selection of target cells and treatment targets is critical. Due to uncertain therapeutic safety and effectiveness, IL-10 signaling blockades remaina region for explorationand are being used in cancer immunotherapy [223]. The effect of a specific blockade of IL-10R on the CD4 + T cell response against hepatitis C virus antigens was evaluated in an ex vivo clinical trial [224]. The results showed that inhibiting IL-10R could induce the proliferation of CD4 + T cells and lead to increased development of interferon-gamma (IFN-γ) against the hepatitis C virus [224]. In the case of immunostimulatory signaling of TLR9 in immune cells modulating the tumor microenvironment in cancer patients, recent studies indicate the involvement of several steps of negative regulation [225]. Therefore, in conjunction with techniques that target immune control point regulation, the CpG ODN-based technique has a great advantage. The latest clinical trials of CpG ODNs in combination with immune checkpoint inhibitors have potential for future cancer immunotherapy, which is effective and comparatively safe [226]. The role of the microbiome in the efficacy of CpG ODN-based treatment still needs to be ascertained.

### 3.6. Microbiome Implications in Adoptive Cell Therapy (ACT)

ACT is usually referred to as the alteration and extension of *in vitro*-induced cancer–cognate lymphocyte infusion to enhance immune function [227]. Tumor infiltrating lymphocytes (TILs), bispecific T-cell engagers (BiTEs), and chimeric antigen receptor (CAR) T-cells [228] are promising approaches for cancer treatment. The isolation accompanied by the ex vivo expansion of tumor-specific T-cells induces ACT. To eliminate cancer, these cells are now transfused back to the patient. CAR T-cells are autologous T-cells that have been previously engineered and redirected to a tumor-specific antigen, in order to enhance antitumor immune response [229]. The link between ACT efficacy and gut microbiome therapy was first demonstrated by Herranz et al. ACT has been found to be successful in HAR mice receiving ACT and in almost completely suppressing tumor growth. Assessment of fecal matter bacteria has shown that, compared to JAX mice, HAR mice have a more diverse variety of *Bacteroidetes* [230]. The level of gram-positive bacteria and gram-negative bacteria intervention widely affected the result of ACT. ACT efficacy improved when the bacterial composition was significantly modified after the decrease inGram-positive bacteria resulting fromintervention with antibiotics [231]. In conclusion, intestinal microbiota plays a vital function in ACT’s antitumor efficacy.

### 3.7. Microbiome Implications in Fecal Microbiota Transplantation (FMT)

Fecal microbiota transplantation (FMT) has been successfully implemented in patients with *C*. *difficile* infection (CDI) and consequential infacilitating clinical improvements andthe restoration of eubiosisand seems to be superior to standard antibiotic therapy [232]. FMT is a kind of infusion of a solution, created with microbiota fecal matter obtained from a donor, into the recipient’s intestinal tract. This directly helps alter the recipient’s microbial composition and impacts health [233]. Metagenomics analysis revealed that there is a prominent reduction indiversity and richness in the case of gut microbiome related to CDI patients, as compared with healthy persons [234]. This dysbiosis process is also characterized by the enhancement of *Proteobacteria* species anda decrement in *Bacteroidetes* and *Firmicutes* species [235]. Thesuccess of FMT is supported by the restored communities of *Bacteroidetes* and *Firmicutes* followed by a decrement in *Proteobacteria* that outcompetes *C*. *difficile*. Nowadays, FMT has been used in clinical trials for additional diseases such as cirrhosis and NASH124 [236].

### 3.8. Microbiome—Host Crosstalk, Signalling and Immunomodulation

The gut is bidirectionally linked with the central nervous system through well-known “gut-brain axis” (GBA), which includes central nervous system (CNS), the autonomic nervous system (ANS), the entero-endocrine system (EES), the enteric nervous system (ENS), and the hypothalamic pituitary adrenal (HPA) axis. Hormones and neuro-hormones secreted helps in cross talk over GBA and modulate the metabolic activities and gastro-intestinal digestive [237]. For that cause, the gut work as complex interface between the gastro-intestinal microbiota and the human body. There is a bidirectional communication system working between gut microbes and host’s GBA, in this system gut work as the communication gatekeeper [238]. A host’s hormones and neuro-hormones can modify the basic composition of gut microbiome during stress response [239]. The gastro-intestinal entero-endocrine cells secrete over 30 different peptide/molecules /hormones (histidine to histamine or glutamate to γ-aminobutyric acid (GABA), short-chain fatty acids (SCFAs), vitamin K etc) involved in a number of functions, like neuromodulation motility, digestive functions and gastro-intestinal [240]. There are a number of signaling pathways that may affect gut microbiome and its metabolites that could perturb the normal physiological functions of host. Interestingly, a number of these processes are controlled by the mammalian target of rapamycin (mTOR) [241]. The mTOR pathway is well-known and is involved in many intracellular processes like transcription, translation, cell growth, cytoskeletal organization autophagy and environmental changes [242]. Crosstalk between the gut microbiota and the mTOR pathway impacts the body’s homeostasis, thus, leading to undesirable complications, not only in cancer, but also in a number of other diseases, like obesity, diabetes, colon and pancreatic immune system malfunctioning and ageing. There are a limited number of research studies regarding the communication between gut microbiota and the mTOR pathway, that elucidate mTOR signaling in microbiota-associated metabolic and immune regulations [243]. Gut microbiome also plays a significant role in the immune system in controlling the functionality and development of gut-associated lymphoid tissues (GALT), including mesenteric lymph nodes, isolated lymphoid follicles, and Peyer’s patches [244,245]. Apart from this gut microbiome secreted products are very important for the immune system to differentiate self from nonself (invaders) at very young stage and activation and maintenance of innate hematolymphoid cells (ILC1, 2, and 3), cytotoxic and noncytotoxic and helper lymphoid cells and natural killer (NK) cells [246]. Natural killer cells and ILC1 produce large quantity of IFN-*γ*, Reg III*γ*, defensins antimicrobial peptides (AMPs), granulysin, and lysozyme, that together play very crucial role in regulation of immune surveillance and microbial ecology [247]. Microbiota also produces vast number of epigenetically active metabolites, like folate, pyridoxine (B_6_), folate (B_9_), cobalamin (B_12_)) A and B vitamins (including riboflavin (B_2_), niacin (B_3_), and pantothenic acid (B_5_), that participate in regulation the genetic responses to environmental signalsand activity of host chromatin-modulating enzymes [248]. Acetyl-CoA formed in number of metabolic processes work as acetyl donor for histone modification (acetylation and deacetylation) that is catalyzed by enzyme histone acetyltransferases. Methionine glycine and serine are substrates for DNA methylation and demethylation enzymes [249,250].

### 3.9. Microbiome and Cancer Prevention

A number of microbiomes-derived molecules participate in anti-tumor activity. Anti-cancer therapies are designed for effective eradication of the cancer control and prevention. As available anti-cancer treatment has been proven to toxic also towards normal cells, microbiomes use may be effective to combat with malignancy.

Microbial-derived SCFAs have quite effective anti-cancer property. For instance, gut bacterial butyrate and propionate inhibit host’s tumor cells histone deacetylases with a general anti-cancer effect. Such a mechanism of anti-tumoral in vitro and in vivo effect of butyrate is well-established in case of both colorectal cancer (CRC) and lymphoma [251,252]. Widely studied bacterial lipopolysaccharide (LPS), a major factor of the outer membrane in gram-negative bacteria, helps in activation of the host’s cell surface receptor toll-like receptor 4 (TLR4), thus in turn activating immune T cell-mediated response against a number of cancer cells [162]. Similarly, the monophosphoryl lipid A (MPL) secreted from *Salmonella enterica* has been widely used as adjuvant along with vaccine against anti-cervical carcinoma [253]. Furthermore, bacterial derived pyridoxine, a group B vitamin, helps in modulation of host’s antitumoral immunosurveillance [254]. Ferricrome metabolite secreted from *Lactobacillus casei*, able to induce apoptosis in tumor cells through JNK pathway activation [255]. *Lactobacilli* may stimulate host’s immune cells such as dendritic cells (DC) or TH1 response and NK cells that leads to the elimination of cancerous cells. Heat-inactivated microorganisms (*Streptococci*) were injected intratumorally for in clinical trial in humans to combat cancer [256,257]. Moreover, Mycobacterium bovis been successfully implicated into bladder of patients, to cure bladder tumor [258,259]. Similarly, Oral administration of *Lactobacillus casei* led to decrement of superficial bladder cancer recurrence [260,261,262,263]. An underlying mechanism involves the direct bacterial stimulation of host’s macrophages and NK cells that triggers strong antitumoral immune response [264]. Intradermal injection of Mycobacterium implicated in case of melanoma and in pancreatic. Ductal carcinoma induces antitumoral immune based response, acting on host’s cytotoxic T cells and antigen presenting cells (APCs) [264,265,266]. Anti-tumoral immune response and also have a direct cytotoxic effect on the tumor cells, by administration of attenuated and/or genetically modified *Salmonella typhimurium* [267,268,269].

## 4. Summary and Concluding Remarks

Although a series of recent studies has been performed on gut microbiome that affects cancer immunotherapy, still much remains unexplored. The mode of action of microbial species that modulates the betterment of immune responses still needs to be understood. Microbes are indispensable in the human gut and for various physiological functions of the human body. Human microbiomes vary among individuals. Various environmental conditions also define their role in the gut. These microbes have a direct impact on various metabolic reactions inside the cell and thus influence the physiology of the cell and cancer treatment by radiation, chemotherapy, immunotherapy, or probiotics. More efforts are required to establish role of each microbes or groups of microbes in different kinds of cancer. Physiological responses to immunotherapy, antibiotic, radiation, and chemotherapy in microbes need to be explored. The gut microbiomecan work as a useful analytical biomarker in immunotherapy. They could be easily manipulated or changed to improve the efficacy of immunotherapy or to minimize side effects during treatment. Basic and multifactorial research can lead to potential treatmentsforcommon cancers. Multidimensional, microbial, molecular, and immunological studies for each kind of cancer need to be conducted. This could help lead to cures, and thus, the realization of the potential of precision and personalized medicine. At present, the enhancement of immune system activity in the diagnosis, cure, and treatment of cancer is of urgent interest. There are several available immunotherapy strategies that involve the manipulation of diverse pathways or molecules. Recently, the consideration of gut microbiomes in immunotherapy has shown a predominated impact on clinical therapeutics. The immunological status of the host, tumor invasion status and biology of the malignancy work as determining factors for individualized therapy. In future, we anticipate that the implementation of a commensal microbiotomewill be a game changer affecting every facet of medical and clinical studies. Analysis of the basic composition of the gut microbiome could lead to accurate evaluations of cancer patients’ healthand will probably help in predicting immune responses and their related adverse effects during cancer therapy. Conclusively, we can say that microbiome explorations might lead to the early treatment of various cancers. However, more research is needed globally to reach conclusions about microbes and establish possible treatments.

## Figures and Tables

**Figure 1 molecules-26-00206-f001:**
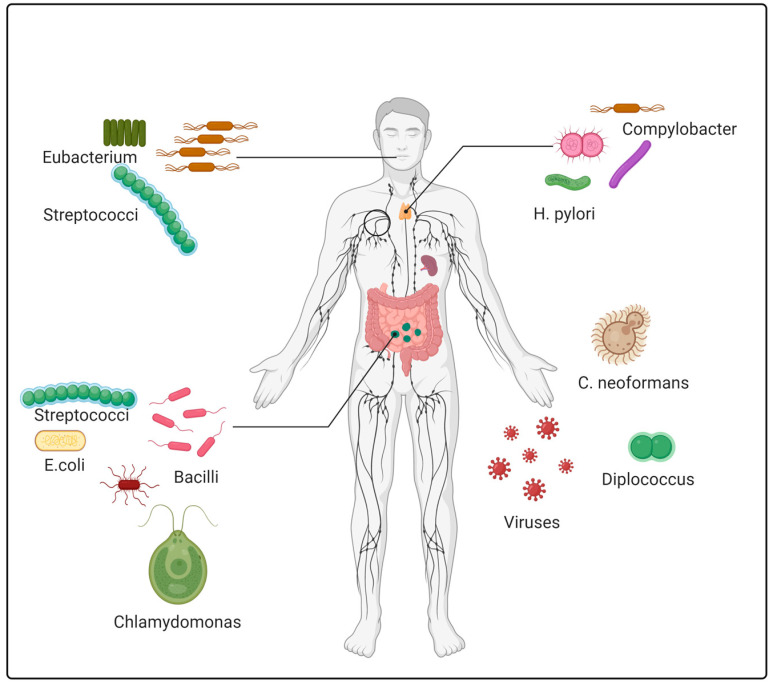
Human microbiomes of various organs. The human gut is a diverse microbial ecosystem comprising viruses, fungi, archaea, and bacteria. These microbiomes affect cancer progression and inhibition in response to various treatments given to patients.

**Figure 2 molecules-26-00206-f002:**
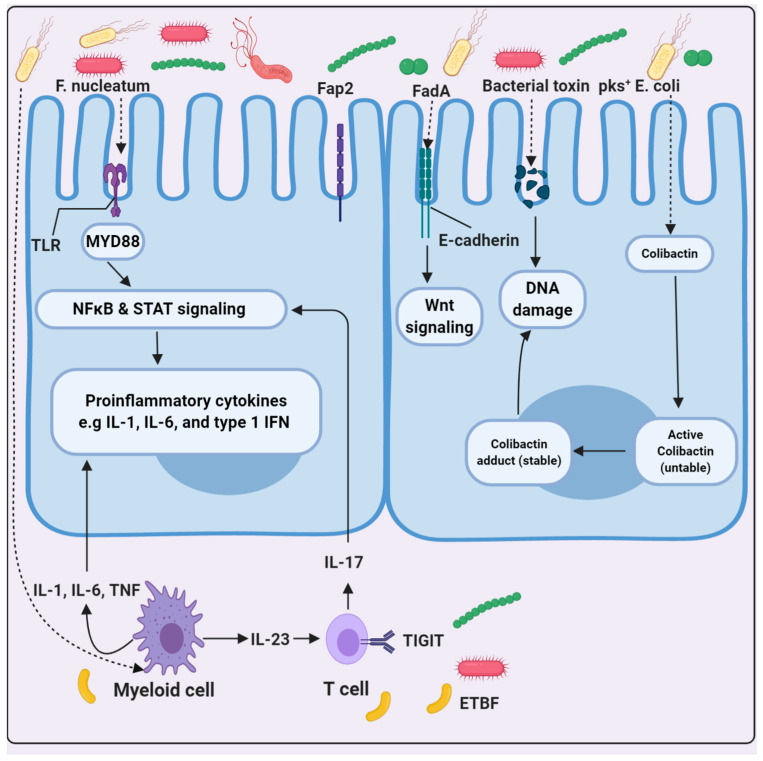
Three major examples of the possible mechanism through which microorganisms cause colorectal cancer. *F. nucleatum*, ETBF, and polyketide synthase-expressing (pks^+^) *E. coli*have a causative role in mediating colorectal cancer.

**Figure 3 molecules-26-00206-f003:**
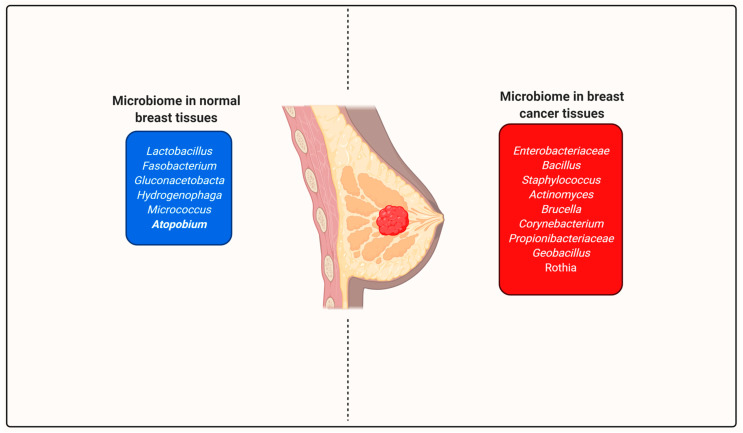
Microbiome changes associated with breast cancer and normal breast tissues; the microbiota mixtures in malignant and benign breast tissue specimens are substantially different and diversity of microbiota is likely to result in carcinogenesis.

**Figure 4 molecules-26-00206-f004:**
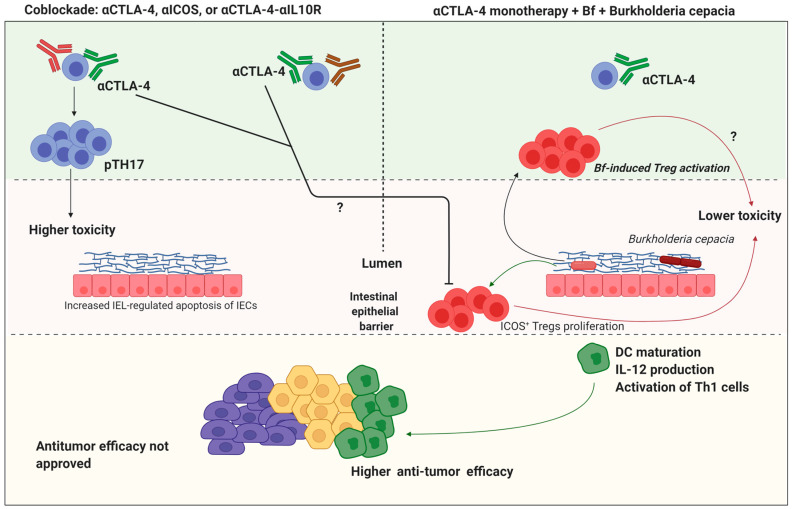
The extent of efficacy and toxicity during the CTLA-4 checkpoint blockade is determined by certain gut microbiotaprofiles.

**Figure 5 molecules-26-00206-f005:**
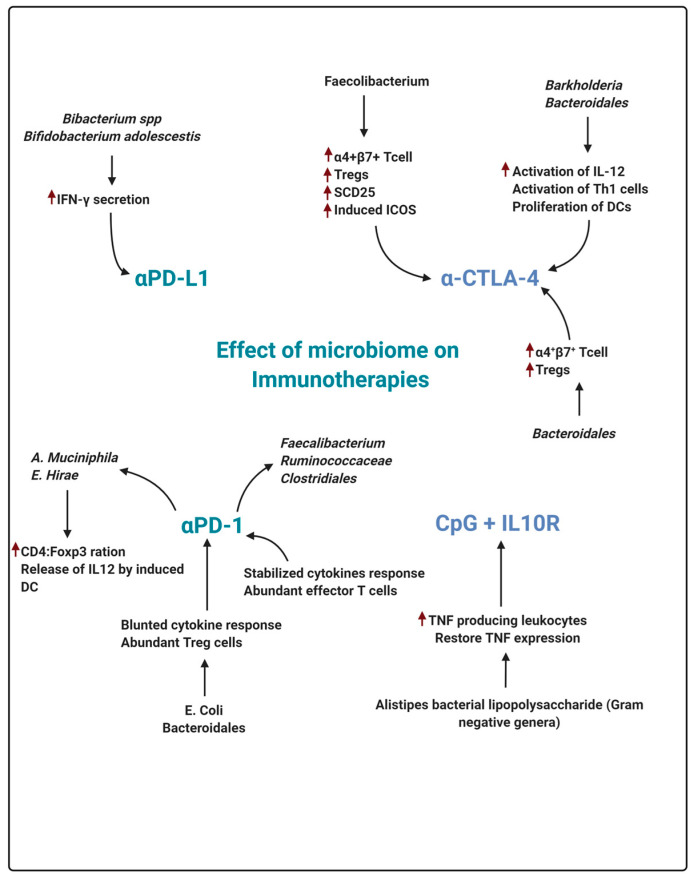
In recent research, unique species of microbiome influences the immune response to four major immunotherapies commonly used and its mechanism. PD-1 programmed death receptor-1, PD-L1: programmed death-ligand 1; CTLA-4: cytotoxic T lymphocyte-associated protein 4; alpPD-1: anti-PD-1 therapy; alpPD-L1: anti-PD-L1 therapy; alpCTLA-4: anti-CTLA-4 therapy; CpG+ alpIL-10R; TLR9 ligand CpG plus anti-IL-10R antibody and various immunotherapy components.

**Table 1 molecules-26-00206-t001:** Microbiota changes reported in human cancer types.

Type of Cancer	Sampling Location	Microbial Increase/Decrease	References
Colorectal	Biopsied tissue and feces materials	Increased: *Escherichia coli*, *Staphylococcus bovis*, *Fusobacterium nucleatum*, *Clostridium* spp., *Streptococcus* spp., and *Bacteroides*. Decreased: Butyrate-producing bacteria, *Lactobacillus*, *Microbacterium*, *Anoxybacillus*, and *Akkermansia muciniphila.*	[26,27,28]
Gall bladder	Bile samples	Increased: *Salmonella paratyphi* and *S. typhi;* Bile is typically considered to be bacteria-free but is infected in many cases.	[28,29]
Esophageal and Barrett’s esophagus	Saliva sample and biopsy tissue	Increased: *S. anginosus*, *Treponema denticola*, *Campylobacter concisus*, *C. rectus* and *S. mitis*. Decreased: *Helicobacter pylori.*	[30,31,32]
Mouth carcinoma	Saliva culture	Increased: *Eubacterium sabureum*, *Leptotrichia buccalis*, *C. ochracea*, *Capnocytophaga gingivalis* and *Streptococcus mitis*	[33,34,35]

**Table 2 molecules-26-00206-t002:** Specific microbes (virus/bacteria) identified to cause various human cancers.

Microbes	Type(s) of Cancer
Human herpes virus 8	Kaposi’ssarcoma
Type 1 human T-cell lymphotropic virus (HTLV-1)	T-cell lymphoma, leukemia (adult)
Hb-B virus	Hepatocellular carcinoma
Hb-Cvirus	Lymphoma, hepatocellular carcinoma
HIV (human immunodeficiency virus)	Kaposi’s sarcoma, lymphomas
EBV (Epstein–Barr virus)	Nasopharyngeal carcinoma, lymphomas
HPV (human papilloma virus)	Oropharyngeal carcinoma, anogenital carcinomas
*Helicobacter pylori*	Esophageal adenocarcinoma, gastric lymphoma, gastric adenocarcinoma

**Table 3 molecules-26-00206-t003:** Several studies on murine models showing tumor-promoting effects of bacterial microbiota.

Cancer	Study Model	Outcome/s	References
Murine Studies			
Breast	Germ-free rats treated with DMAB.	Reduced tumors growth in germ-free rats.	[50]
Lung	Germ-free rats treated with NHMI.	Fewer tumors in male germ-free rats. No observed changes in female germ-free rats.	[51]
Gastric	1. INS-GAS mice (gnotobiotic) infected with *Helicobacter pylori.*	Fewer tumors in germ-free mice.	[52]
	2. Antibiotic-treated INS-GAS mice infected with *Helicobacter pylori*.	Fewer tumors in mice treated with antibiotics.	[53]
Liver	1. Germ-free mice treated with (DEN) and CCl_4_.	Fewer tumors in germ-free mice.	[53]
	2. An antibiotic cocktail was administered to DEN and CCl_4_-treated mice.	Fewer tumors in antibiotic-treated mice.	[54]
	3. Rifaximin administered to DEN and CCl_4_-treated mice.	Fewer tumors in rifaximin-treated mice.	[54]
	4. Neomycin administeredto DEN-treated rats.	Fewer tumors in neomycin-treated rats.	[54]
	5. Vancomycin administeredto DMBA-treated miceon a high-fatdiet.	Fewer tumors in vancomycin-treated mice.	[55]
	6. An antibiotic cocktail administered to DMBA-treated mice on a high-fat diet.	Fewer tumors in antibiotic-treated mice.	[55]
Colorectal	1. Germ-free mice (*Apc*Min/+).	Fewer tumors in germ-free mice.	[56]
	2. Gnotobiotic mice (AOM in *IL-10*−/−).	Fewer tumors in germ-free mice.	[57]
	3. Mice (*Apc*Min/+ *Cdx2*-Cre) treated with an antibiotic mixture.	Fewer tumors in antibiotic-treated mice.	[58]
	4. Mice *(Nod1*−/−) treated with an antibiotic mixture.	Fewer tumors in antibiotic-treated mice.	[59]
	5. Mice (DSS and AOM) treated with an antibiotic mixture.	Fewer tumors in antibiotic-treated mice.	[60]
	6. Mice (*Nod2*−/−) were transplanted with wild-type microbiota.	Fewer tumors after transplant.	[60]

**Table 4 molecules-26-00206-t004:** Microbes and carcinogenic mechanisms.

Type of Cancer	Role of Microbes/Mechanism of Carcinogenesis	Evidence/Proof	References
Gastric lymphoma of the gastric MALT, IPSID, MALT lymphoma of the skin, adnexal ocular lymphoma	Chronic infection with *Helicobacter pylori* Patients with chronic infection with *H. pylori*, *Campylobacter jejuni*, *Borreliaburgdorferi*, or *Chlamydia psittaci*	Epidemiology support Reduction by *H. pylori* Eradication Antibiotic treatment	[63,64,65,66,67,68,69,70,71,72,73]
Esophageal	Decreased risk in patients who have *H. pylori* infection	Epidemiology support	[70,74]
Gallbladder	Chronic infection with *Salmonella enteric* subsp. *enterica* serovar *Typhi*	Epidemiology support	[75,76]
Breast	Increased T regulatory cell-mediated inflammation	Cancer promoted in *Apc*Min/+ mice infected with *Helicobacter hepaticus*	[77]
Liver	Chronic hepatitis	Cancer facilitated in mice infected with *H. hepaticus*	[78]
Colorectal	TNF-mediated and NO-mediated	Cancer supported in *Rag2*^−/−^ mice infected with *H. hepaticus*	[79]
Colorectal	Barrier failure Dysbiosis Bacterial genotoxicity Chronic inflammation	Cancer reduction by antibiotics and in germ-free mice; transmission of dysbiotic microbiota triggers cancer development	[80,81,82,83,84]
Liver	Increased liver sensitivity to MAMP-activating TLRs Increased sensitivity to the secondary bile acid (DCA)	Cancer reduction by treatment with antibiotics and in germ-free mice Cancer increased by treatment with LPS and DCA	[85,86]
Lung	Increased bacterial infection in COPD	Decreased cancer in germ-free animals The promotion of cancer by LPS and infections	[87,88,89,90,91]
Pancreatic	LPS-TLR4-mediated increase	LPS treatment increases cancer development	[92,93,94]

**Table 5 molecules-26-00206-t005:** Unique clinical trials carried out in response to cancer therapy worldwide to improve the gut microbiome [159].

Global NCT Number	Type of Malignancy	Aim	Intervention	Results/Outcome Measures	Place
03290651	Breast cancer	To determine if oral antibiotics can change the breast flora	Probiotics Natural Health Product-RepHresh Pro-B	Change in breast microbiota, inflammatory markers	Canada
03341143	Melanoma	To establish concurrent use of FMT and pembrolizumab in patients with PD-1-resistant melanoma	FMT (donor responder to PD-1 therapy) with pembrolizumab	ORR, change in T cell composition and function, change in innate and adaptive immune subsets	USA
00936572	Colorectal cancer	To investigate the effect of probiotics on gut microflora and the immune and inflammatory response	Probiotics (La1, BB536)	To perform morphological and microbiological evaluation of the colonic microflora, GI function	Italy
03072641	Colorectal cancer	To reactivate the tumor-suppressor genes using probiotics	ProBionClinica (*Bifidobacteriumlactis*, *L*. *acidophilus*)	Changes in microbiota composition and DNA methylation	Sweden
01609660	Colorectal cancer	To assess the impact of probiotics on patients undergoing colorectal resections	*Saccharomyces boulardii*	To measure mucosal cytokines, SCFA postoperative complications, and hospital LOS	Brazil
00197873	Colorectal cancer	To prevent chemotherapy-induced diarrhea	*L*. *rhamnosus* supplementation	Effect on treatment-related toxicity other than diarrhea	Finland
02269150	Malignancies requiring allo-HSCT	To assess the utility of FMT in prevention of CDI in patients who underwent allo-HSCT	Auto-FMT	CDI	USA
02928523	Acute myeloid leukemia	To use FMT to prevent complications associated with dysbiosis in patients undergoing intensive treatment	Auto-FMT	Dysbiosis correction, eradication of multidrug resistant bacteria, definition of dysbiosis, biosignature	France
03552458	Head-and-neck cancer	To assess the role of probiotics in preventing oral mucositis	*Lactobacillus reuteri* Oral Solution (BioGaia)	Oral mucositis severity, oral bacterial genetics, and transcriptional analysis	Singapore

**Table 6 molecules-26-00206-t006:** Manipulation ofthe gut microbiome to alleviate responses to cancer immunotherapy [160].

Accession or Trial Number	Intervention	Targeted Patient Population	Finding(s)
NCT03072641	Irregular probiotics supplements (ProBionClinica*B*. *lactis* BI-04, *L*. *acidophilus* Inulin+ NCFM) Intake	CRC patients ages18^+^	Primary: change in fecal and tumor microbiota. Secondary: Changes in epigenetics patterns of tumor tissue.
NCT01895530	Randomized probiotic (*S*. *Boulardii)* administration	CRC patients ages 18^+^ undergoing elective CRC resection	Primary: cytokine expression in colonic mucosa (via qPCR). Secondary: post-operative complications.
NCT03358511	Single-arm probiotics (Primal Defense Ultra multi-strain probiotics formula)	Post-menopausal breast cancer patients (stages I–III)	Primary: change in mean number of CD8^+^ cells.
NCT03353402	Single-arm FMT (colonoscopy or gastroscopy) from patient donors who responded to immunotherapy	Metastatic melanoma patients ages 18^+^ who previously failed standard therapies	Primary: safety (AEs associated with FMT), engraftment of FMT. Secondary: changes in immune cell populations and activity, objective response rate.
NCT02928523	Single-arm autologous FMT (frozen inoculum)	Acute myeloid leukemia patients ages 18–65 treated with intensive chemotherapy and antibiotics	Primary: diversity of the gut microbiome, multi-drug-resistant bacteria eradication. Secondary: signature of dysbiosis. of the gut microbiome.
NCT02079662	Randomized intensive lifestyle change (diet, exercise, psychosocial)	Stages II and III breast cancer patients treated at MDACC ages 18^+^	Primary: disease-free survival (DFS). Secondary: change in fecal and oral microbiome (via 16S profiling).
NCT02843425	Addition of ½ cup beans per day to regular diet in a crossover design	All cancer patients treated at MDACC	Primary: change in fecal microbiome profile.

**Table 7 molecules-26-00206-t007:** Link between the compositions of the gut microbiome and the effectiveness of carcinoma therapy.

Main Outcome	Data Source	Carcinoma/Treatment	References
Immunological therapy Commensal type of microbiota needed for standard response to treatment	Rodent—mouse	Several models of carcinoma/anti-IL-10R antibody + CpG oligonucleotide and oxaliplatin (platinum-chemotherapy)	[161]
Irradiation of whole body with disruption of the intestine barrier and enhanced response of T-lymphocyte cell-mediated treatment via mechanisms dependent upon TLR4 signaling/microbe translocation/LPS	Rodent—mouse	Melanoma/Adoptive T-lymphocyte transfer of cells	[162]
*Firmicutes* and *Faecalibacterium* presence in baseline samples of stool liked with ICB response; abundance of *Bacteroides* linked with low ICB response	Human	Melanoma (Metastatic stage)/CTLA-4 inhibitor	[163]
Abundance of *Blautia* linked with overall increased survival rate and decreased GVHD risk	Human	Hematologic cancers/Allo-HSCT	[164]
*Bacteroides* abundance was associated with resistance to ICB-induced colitis	Human	Melanoma (Metastatic stage)/CTLA-4 inhibitor	[165]
Abundance of *Bifidobacterium* linked with enhanced automatic immunity against tumor and ICB response	Rodent—mouse	Melanoma/Anti-PD-L1 inhibitor	[166]
Abundance of *Bacteroides* liked with ICB response	Rodent—human	Melanoma (Metastatic stage)/CTLA-4 inhibitor	[167]
Abundance of *Eubacterium limosum* linked with reduced chances of progression of disease or relapse	Rodent—human	Blood cancer/Allo-HSCT	[168]
*Faecalibacterium prausnitzii*, *Bacteroides caccae*, *Holdemaniafiliformis*, *Bacteroides thetaiotaomicron*, and *Doreaformicigenerans* linked with ICB response	Human	Melanoma (Metastatic stage)/CTLA-4 inhibitor; PD-1 inhibitor	[169]
Large number of bacteria in the sample of baseline stool found to be enriched differentially between strong ICB response patients vs. poor ICB response patients	Human	Mouse-melanoma (Metastatic)/PD-1 inhibitor	[170]
Clostridiales, high microbiome richness and abundance of *Faecalibacterium*, Ruminococcaceae with ICB response in baseline stool samples	Mouse; Human	Melanoma (Metastatic stage)/PD-1 inhibitor	[171]
Abundance of *A*. *muciniphila* in the samples of baseline stool found to be linked with ICB response	Mouse; Human	lung carcinoma of non-small cell; Renal cancer/PD-1 inhibitor	[172]
Chemotherapy with immunostimulatory properties Existence of intratumoral Gammaproteobacteria linked with gemcitabine chemotherapy resistance	Mouse; Human	Adenocarcinoma (Ductal Pancreatic)/Gemcitabine immunomodulatory chemotherapy treatment	[173]
Abundance of *Akkermansia muciniphila* in the samples of baseline stool found to be liked with ICB response	Rodent-Mouse	Several models of carcinoma/immunomodulatory chemotherapy Cyclophosphamide	[174]

## Data Availability

No new data were created or analyzed in this study. Data sharing is not applicable to this article.

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
