# Peer review of "The Microbiome and Its Implications in Cancer Immunotherapy"

_molecules, 2021, doi:10.3390/molecules26010206_

Round 1

Reviewer 1 Report

This review aims to address the role of the microbiome in cancer immunotherapy.

Generally this is a poor quality manuscript, the writing is convoluted, there are many factual inaccuracies and the references frequently do not support the statements they are attached to.

Major Issues

  1. The manuscript frequently conflates correlation and causation and this needs to be corrected.
  2. There are many inaccurate statements from the abstract onwards eg line 12 'immunotherapy, chemotherapy and more recently radiotherapy' - radiotherapy is a long standing treatment, it is immunotherapy that is more recent. eg line '36 major microbes are found in the mouth and stomach' - major microbiome communities are found in the intestinal tract and colon, on the skin and in the vagina.
  3. References are inappropriate - eg line 45-46 'can initiate a number of human metabolic disorders, including cancer [6]' Reference 6 looks at poultry not humans. Generally it is possible that the bibliography was not updated prior to submission as many references are incorrect.
  4. S.No (study number?) seems irrelevant in tables  instead there should be a column for the actual reference/references for the studies described.
  5. It is unclear how the researcher has selected the studies addressed as not all relevant studies are included and the fact that findings differ between different studies re involved microbes and direction of alteration is not addressed.
  6. It is unclear in many places where human vs animal studies are being discussed, frequently animal studies are cited to support statements made about human disease.
  7. The writing is very convoluted, has multiple internal inconsistencies and needs extensive editing and re-arrangement.

Author Response

Thank you so much for reviewing the article and appreciate your insights for improving it. All the suggested comments are incorporated in the revised manuscript. 

Comment 1. The manuscript frequently conflates correlation and causation and this need to be corrected

Response: I have rewritten some sections to incorporate your suggestion in the article.

Comment 2. There are many inaccurate statements from the abstract onwards eg line 12 'immunotherapy, chemotherapy and more recently radiotherapy' - radiotherapy is a long standing treatment, it is immunotherapy that is more recent. eg line '36 major microbes are found in the mouth and stomach' - major microbiome communities are found in the intestinal tract and colon, on the skin and in the vagina.

Response: I have corrected in respective line. The suggestion has been incorporated and highlighted with yellow in the manuscript.

Comment 3. References are inappropriate - eg line 45-46 'can initiate a number of human metabolic disorders, including cancer [6]' Reference 6 looks at poultry not humans. Generally it is possible that the bibliography was not updated prior to submission as many references are incorrect.

Response: I have corrected in respective line. All the references have been updated, corrected and highlighted with yellow in the manuscript.

Comment 4. S.No (study number?) seems irrelevant in tables instead there should be a column for the actual reference/references for the studies described.

Response: As suggested, I have added the references in the tables.

Comment 5. It is unclear how the researcher has selected the studies addressed as not all relevant studies are included and the fact that findings differ between different studies re involved microbes and direction of alteration is not addressed.

Response: I have updated the manuscript with recent studies. These were selected based on novelty and relevance to cancer and microbiome.   

Comment 6  It is unclear in many places where human vs animal studies are being discussed, frequently animal studies are cited to support statements made about human disease.

Response: Germ-free (GF) mice are a relevant model system to study host–microbial interactions in health and disease. In order to establish the role of microbes in certain types of human diseases through host and microbe’s interaction, citation of ongoing research on germ-free mice and knockout mice related to microbiome are important. 

Comment 7.   The writing is very convoluted, has multiple internal inconsistencies and needs extensive editing and re-arrangement?

Response: I have rewritten those paragraphs and lines of manuscript to avoid internal inconsistencies highlighted with yellow. Additionally, the manuscript was edited using the MDPI’s specialist English editing services. 

Reviewer 2 Report

The authors have collected and discussed reports that show impact of the microbiome on cancer  progression and resistance to cancer therapy and immunotherapy.

However, the concept of microbiome presented in this manuscript is misleading. The microbiome are all the  bacteria, virus, protozoa and funghi that live in or on the body, but are NOT PATHOGENS. A change in their composition or number can contribute  to immunosuppressive or pro-inflammatory conditions  and exacerbate inflammatory diseases , including cancer. However, the authors  show tables of viral pathogens, as HIV, HTLV, that are acquired pathogens, and are not part of the microbiome. Yes, the microbiome has a critical effect in facilitating infection and disease progression by these pathogens, i.e HIV. This has to be clarified and discussed.

The authors do not discuss any signaling pathways involved in modifying the immune response to alterations in the microbiome, i.e. modifying immunometabolism and mTOR signaling The authors should discuss some of the most novel works in this field.

The authors do not discuss risk factors that lead to  changes in the microbiome and how management of those risk factors might  contribute to cancer prevention.

First clinical cancer trials conducted so far to modify the microbiome using antibiotics in combination with ICs have not shown a clear survival advantage. Also novel biomarker associated to the microbiome capable to predict disease progression  and  outcome to  therapy have not  been identified to date.

Author Response

I am very thankful for your valuable suggestions to enhance the manuscript. All the comments are incorporated in the revised manuscript.

Comment 1. However, the concept of microbiome presented in this manuscript is misleading. The microbiome are all the bacteria, virus, protozoa and fungi that live in or on the body but are NOT PATHOGENS. A change in their composition or number can contribute  to immunosuppressive or pro-inflammatory conditions  and exacerbate inflammatory diseases , including cancer. However, the authors show tables of viral pathogens, as HIV, HTLV, that are acquired pathogens, and are not part of the microbiome. Yes, the microbiome has a critical effect in facilitating infection and disease progression by these pathogens, i.e HIV. This has to be clarified and discussed.

Response: I have corrected in respective paragraph. The suggestion has been incorporated and highlighted with yellow within the manuscript. As you have mentioned the aim is to highlight that microbiome has a critical impact on facilitating infection and cancer progression by these microbial pathogens.

Comment 2. The authors do not discuss any signaling pathways involved in modifying the immune response to alterations in the microbiome, i.e., modifying immunometabolism and mTOR signaling the authors should discuss some of the most novel works in this field.

Response: I have corrected and added your valuable suggestion at the end of the manuscript. The suggestion has been incorporated and highlighted with yellow in the manuscript.

Comment 3. The authors do not discuss risk factors that lead to changes in the microbiome and how management of those risk factors might contribute to cancer prevention

Response: I have corrected and added your valuable suggestion at the end of the manuscript. The suggestion has been incorporated and highlighted with yellow in the manuscript.

Comment 4. First clinical cancer trials conducted so far to modify the microbiome using antibiotics in combination with ICs have not shown a clear survival advantage. Also novel biomarker associated to the microbiome capable to predict disease progression and  outcome to  therapy have not  been identified to date.

Response: I have incorporated your suggestion in the manuscript.

Reviewer 3 Report

The author described the influence of microbiome on anticancer immunotherapy.

The article is well written and interesting but there are many weakness

Instructions for Authors: The abstract should be a total of about 200 words maximum. Now abstract is 276 words.

34-35 line. Which vitamin are production by microbiome? Which acids?

Table 1 and Table 2. Helicobacter pylori causes Esophageal adenocarcinoma (table 2). In Esophageal adenocarcinoma patients is decrease in Helicobacter pylori (table 1).

Please add bibliographic references to each line in all tables.

117 line. “lead to inflammation via IL-10 generation.” IL-10 is anti-inflammation cytokine.

510,572,574 line “in vitro”, “in vivo”, “ex vivo” should be italicized

557,430 line „IL10” should be “IL-10”

A list of abbreviations should be added at the end of the article

Author Response

Thank you so much for kind and valuable suggestions. All the comments and suggestions are incorporated in the revised manuscript.

Comment 1. The abstract should be a total of about 200 words maximum. Now abstract is 276 words.

Response: I have incorporated your suggestion in the manuscript.

Comment 2. 34-35 line. Which vitamin is production by microbiome? Which acids?

Response: Correction has been done in the revised manuscript.

Comment 3. Table 1 and Table 2. Helicobacter pylori causes Esophageal adenocarcinoma (table 2). In Esophageal adenocarcinoma patients is decrease in Helicobacter pylori (table 1).

Response: Correction has been made in the revised manuscript.

Comment 4. Please add bibliographic references to each line in all tables.

Response: References has been incorporated in the tables.

Comment 5. 117 line. “lead to inflammation via IL-10 generation.” IL-10 is anti-inflammation cytokine.

Response: Correction has been done in the revised manuscript.

Comment 6. 510,572,574 line “in vitro”, “in vivo”, “ex vivo” should be italicized

Response: Correction has been done in the revised manuscript.

Comment 6.  557,430 line „IL10” should be “IL-10”

Response: Correction has been done in the revised manuscript.

Comment 7.  A list of abbreviations should be added at the end of the article.

Response: List of abbreviations have been given at the end of the revised manuscript.

Round 2

Reviewer 2 Report

The author has  addressed in detail all my concerns.